# Research on the n-Stage Delay Distribution Method Based on a Compensation Mechanism in a Random Environment

**Lei Zhou** 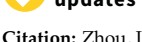, **Yue Qi** and **Fachao Li** *

School of Economics and Management, Hebei University of Science and Technology, Shijiazhuang 050018, China; zhoulei19@126.com (L.Z.); qi15227168996@163.com (Y.Q.)
* Correspondence: lifachao@tsinghua.org.cn

**Abstract:** With the development of logistics, the delayed distribution problem based on a compensation mechanism has appeared. The core of this problem is to decide whether to delay the distribution at the beginning of each stage and how to compensate the customers if delay occurs. In this paper, beginning with the 2 and 3-stage delay distribution problem, the characteristics and computational complexity of the model are analyzed, and a formal model description of the n-stage problem is given. The expected value and variance are used as the centralized quantization description strategy for random variables, and the expected value model and the generalized expectation value model for solving the delay distribution problem are given. The solution algorithm is given, and the dependence of the single transport cost of each transport vehicle and the penalty for each car delay in a period-of-time distribution are analyzed. Combined with specific examples, theoretical analysis and example calculations show that the formal description model is a good platform for further combinations of solution methods. This method extends the general delayed distribution problem to multi-stage delayed distribution, which has guiding significance for decision-makers. The proposed model has solid system structure features and interpretability, and could be used in a wide variety of applications.

**Keywords:** stochastic programming; compensation mechanism; minimize expectation cost; non-load distribution; generalized expectation value

## 1. Introduction

Together with the reform and opening up policies in China, "the third profit source" in enterprise logistics has become increasingly important for economic activity. As an important part of logistics activity, distribution directly affects how consumers evaluate logistics enterprises. Delayed delivery is a common problem in vehicle logistics, which leads to a decline in customer satisfaction and affects secondary consumption, and at the same time makes the relationship between customers and logistics enterprises tense, which is not conducive to the long-term development of logistics enterprises. For this reason, optimization of distribution is necessary. Because all uncertain information processing policies are different, the resulting solutions are often different.

Vehicle load distribution is an important component of logistics optimization. At the core of vehicle load distribution is how to reasonably arrange the goods to minimize the number of vehicles and ensure the vehicle loading rate without exceeding the load capacity of vehicles. Michael et al. [1] analyzed the vehicle logistics with precise methods and optimized the vehicle logistics. Gizem et al. [2] introduced and considered a dynamic variant of the vehicle routing problem with roaming delivery locations and developed an iterative re-optimization framework to solve it and focus on using branch-and-price in a dynamic decision-making environment to investigate its potential as a solution methodology for operational problems. Liu et al. [3] presented a combinatorial optimization model for the courier delivery network design problem, and optimization aims to determine the

transportation organization mode for each courier flow considering the frequency delay of accumulation in the sorting process. Wang et al. [4] proposed the cooperation strategy for the green pickup and delivery problem (GPDP). Some researchers pay more attention to logistics and transportation optimization problems [5–10]. The above literature concerns reasonable optimization of distribution.

Many random factors in distribution can be represented using stochastic programming, and stochastic programming has been well studied. Al-Khamis et al. [11] proposed a two-stage stochastic programming model for the parallel machine scheduling problem in which the objective was to determine the machine capacities that maximized the expected net profit of on-time jobs with uncertain due dates. Noyan et al. [12] considered a risk-averse two-stage stochastic programming model with conditional value-at-risk (CVaR) as the risk measure. This approach constructed two decomposition algorithms based on the generic Benders-decomposition approach to solving such problems. Abdelaziz et al. [13] surveyed various solution approaches for multi-objective stochastic problems in which random variables can exist in both the objectives and the constraint parameters. Abdelaziz et al. [14] established stochastic programming models from different perspectives and proposed their own approaches and methods to solve stochastic problems. Wang et al. [15] focused on finding a priori evacuation plans by considering side constraints, scenario-based stochastic link travel times, and capacities and proposed a stochastic programming framework for the reorganization of traffic routing for disaster response. Zahiri et al. [16] proposed a novel multi-stage probabilistic stochastic programming approach and created a real post-disaster relief distribution planning case study. Goberna et al. [17] dealt with uncertain multi-objective convex programming problems and presented sufficient conditions for the existence of highly robust weakly efficient solutions, that was, robust feasible solutions which were weakly efficient for any possible instance of the objective function within a specified uncertainty set. Ogbe et al. [18] proposed a new cross-decomposition method combining two classical decomposition methods. The method outperformed Benders' decomposition when the number of scenarios was large. Hasany et al. [19] developed a two-stage stochastic program for the railroad blocking problem that considers the uncertainty inherent in demand and supply resource indicators and developed two exact algorithms based on the L-shaped method. Niu et al. [20] presented a $d$-minimal cut-based algorithm to evaluate the performance index $R_{d+1}$ of a distribution network, defined as the probability that a specified demand $d + 1$ can be successfully distributed through stochastic arc capacities from the source to the destination. Zheng et al. [21] developed a multimodal macroscopic fundamental diagram (MFD) to represent the traffic dynamics in a multimodal transport system. Optimization was performed to minimize the total passenger hours traveled (PHT) to serve total demand by redistributing the road space among modes. Yu et al. [22] addressed a new variant of the vehicle routing problem, called the two-echelon vehicle routing problem with time windows, covering options, and occasional drivers. Mancini et al. [23] proposed a mixed delivery approach, which combines attended home delivery and f shared delivery location usage innovatively. Customers can either be served at home during their preferred time window, or they can be asked to pick up their parcel at one of the f shared delivery locations. Wang et al. [24] proposed the two-stage delay distribution method based on a compensation mechanism under a random environment. Zhou et al. [25] proposed the compensation mechanism of delayed distribution based on interest balance. Some researchers pay more attention to the logistics distribution problem [26–37].

There are shortcomings in the above-mentioned literature: Firstly, some researchers proposed the compensation mechanism for the delay distribution, but the research is too scant to improve the development of the delay distribution method. Secondly, there is no delay distribution involving $n$ stages which only research stages 2–3. Furthermore, it is difficult to calculate the algorithm. Only the expectation or probability distribution is considered for the treatment of random variables without considering the variance, while the model error is large. Therefore the n-stage delay distribution method based on a compensation mechanism will be established to solve the above problems.

The remainder of this paper is organized as follows. Section 2 contains the separate section notations. Section 3 reviews the literature. In Section 4, a delay distribution method based on penalty and the choice mechanism is analyzed. The solution procedure is proposed in Section 5. A numerical example is given in Section 6 to illustrate the effectiveness of the proposed model. The demarcation point of the delayed and non-delayed distribution is obtained, and the optimal solution is identified. A conclusion and recommended future research directions are given in the final section.

For narrative convenience, this paper uses the following definitions and assumptions: (1) $\text{int}(x)$ is the integer part of $x$, $x \in \text{R}$, (e.g., $\text{int}(1.1) = 1$); and (2) $\text{mod}(d, b) = d - b\,\text{int}(d/b)$ is the remainder function of $d$ about $b$ (here, $d$ and $b$ are natural numbers, and $b \neq 0$).

## 2. Notations

### 2.1. Indices and Sets

For narrative convenience, the Indices and sets descriprion is shown in Table 1.

**Table 1.** Indices and Sets Description.

| Indices and Sets | Explanation |
|---|---|
| $(\Omega, B, \text{Pr})$ | Is a probability space. |
| $\Omega$ | Is the sample space. |
| $B$ | Is the event space. |
| $d$ | Is the spatial dimension. The value range is a set of natural numbers. |
| $i$ | Is the number of all possible random demand values in the second stage. It is a non-negative integer. |
| $j$ | Is the number of current stages. The value range is a set of natural numbers. |
| $n$ | Is the total number of stages. The value range is a set of natural numbers. |
| $\text{Pr}$ | Is the probability. The value range is $[0, 1]$. |
| R | Is a real space. The value range is $(-\infty, +\infty)$. |
| $\text{R}^d$ | Is the $d$-dimensional real space. It is a The value range is $(-\infty, +\infty)^d$. |

### 2.2. Parameters

The parameters description is shown in Table 2.

**Table 2.** Parameters Description.

| Parameters | Explanation |
|---|---|
| $\xi_j$ | Is the order quantity in the $j$th stage. It is a random variable. The value range is a set of natural numbers. |
| $E(\xi_j)$ | Is the mathematical expectation of the random variable $\xi_j$. |
| $D(\xi_j)$ | Is the variance of the random variable $\xi_j$. |
| $\theta$ | Is the penalty for each car delay in a period-of-time distribution. It is continuous. |
| $c$ | Is the single transport cost of each transport vehicle. It is continuous. |
| $l_j$ | Is all possible values of $\xi_j$. $l_j = 0, 1, 2, \cdots, Q - 1$. |
| $Q$ | Is the single distribution vehicle load capacity. It is an integer. |
| $y_1$ | Is the all possible values of $Z_1(4, \xi_2)$. It is continuous. |
| $y_2$ | Is the all possible values of $Z_2(4, \xi_2)$. It is continuous. |
| $Z_1(k, \xi_2)$ | Is the cost function of choosing delayed delivery in the first stage. It is a random variable function and continuous. |
| $Z_2(k, \xi_2)$ | Is the cost function of choosing delayed delivery in the second stage. It is a random variable function and continuous. |

**Table 2.** *Cont.*

| Parameters | Explanation |
|---|---|
| $Z_{11}(k, \xi_2, \xi_3)$ | Is the distribution cost when we use the delayed distribution in stages 1 and 2. It is a random variable function and continuous. |
| $Z_{12}(k, \xi_2, \xi_3)$ | Is the distribution cost when we use the distribution strategy in which delivery is delayed in stage 1, and no distribution delay occurs in stage 2. It is a random variable function and continuous. |
| $Z_{21}(k, \xi_2, \xi_3)$ | Is the distribution cost when we use the distribution strategy in which delivery is delayed in stage 2, and no distribution delay occurs in stage 1. It is a random variable function and continuous. |
| $Z_{22}(k, \xi_2, \xi_3)$ | Is the distribution costs when we do not use the delay distribution in stages 1 and 2. It is a random variable function and continuous. |

*2.3. Decision Variable*

The decision variables description is in Table 3.

**Table 3.** Decisions Variable Description.

| Indices and Sets | Explanation |
|---|---|
| $k$ | It is the part without delivery. It is an integer. |
| $x_j$ | If we choose delayed delivery on the $j$th delivery stage, $x_j = 0$, the amount of delayed delivery is $k$ ($k < Q$), and each delayed delivery of goods needs a penalty of $\theta$, otherwise $x_j = 1$, $i = 0, 1, 2, \cdots, n - 1$. They are binary. |

**3. Delay Distribution Characteristic Analysis in a Random Environment**

According to vehicle distribution characteristics, customer demand is a random variable. Therefore, the distribution model is a stochastic programming problem. In the distribution process, each nodal transportation task can be divided into non-fully loaded and fully loaded transport. Fully loaded transport can fully use the loading carrier capacity. Non-fully loaded transport is the total transportation task minus the remaining goods of the load transportation task. When the goods reach the conditions for full-load transportation, they need to be delivered on time on the delivery date. If they do not meet the conditions for full-load transportation, an additional cost is required for on-time delivery, which produces a certain penalty for delayed delivery. In this case, an important problem is determining the transportation plan that can minimize the cost. Based on the above situation, building an appropriate random distribution model and selecting the appropriate distribution threshold are highly important tasks for managers to make the correct decisions.

Delay distribution is an added limit, and additional uncertainty coexists in the complex decision problem, which has attracted widespread attention in academic applications. This section considers the full-load distribution problem in a random environment and discusses the method for determining the distribution mechanism.

To improve the service quality and management system, a car manufacturer transfers the car distribution to a third-party logistics company. The relevant regulations are as follows: (1) Set up $n$ delivery days in each order cycle, that is, $n$ time nodes; (2) All orders must be delivered within the order period; (3) If the number of orders on the current delivery day does not meet the conditions of full-load transportation, you can choose to postpone the delivery to the next delivery day, but delayed delivery requires a certain delivery penalty. In real life, based on the customer demand, the manufacturer supplies compensation to the customer, which can be viewed as a manufacturer's penalty. We consider the delay choice in the process of the logistics distribution system.

Because delayed delivery is or is not closely related to the order number for the next time period, and because the next period of orders is a variable that depends on many factors, it needs to be treated as a random variable in data processing, the selection of

delay delivery is a decision-making problem in a random environment. In this paper, we consider using the $n = 2$, $n = 3$, $n = 4$ and $n = 10$ delay distribution selection mechanisms to describe and verify. We make the following assumptions: If the order quantity does not meet the condition of full-load distribution, the distribution can be delayed; otherwise, the distribution cannot be delayed.

*3.1. Choice Mechanism of 2-Stage Delay Distribution*

For the first stage in which we cannot load distribution orders $k$, and if, for the second stage, the order quantity is $\xi_2$ with distribution $\Pr(\xi_2 = l_2) = p_i$, $l_2 = 0, 1, 2, \cdots, Q - 1$. $Z_1(k, \xi_2)$ is the cost function of choosing delayed delivery in the first stage, $Z_2(k, \xi_2)$ is the cost function of choosing delivery in the first stage. In the first stage, $k$ cars are delayed for delivery. Respectively, and these values are random variables; the distributions are as follows:

$$Z_1(k, \xi_2) = k\theta + \text{int}\left(\frac{k + \xi_2}{Q}\right)c + \delta\left(\frac{k + \xi_2}{Q}\right)c,$$

$$Z_2(k, \xi_2) = c + \text{int}\left(\frac{\xi_2}{Q}\right)c + \delta\left(\frac{\xi_2}{Q}\right)c.$$

where $\delta(t) = \begin{cases} 0, & \text{when } t \text{ is an integer,} \\ 1, & \text{other.} \end{cases}$

For example, when $Q = 13$, $k = 4$, we consider the delay decision method. Using probability theory, we obtain the distribution of $Z_1(4, \xi_2)$ and $Z_2(4, \xi_2)$ as shown in Tables 4 and 5, respectively.

**Table 4.** The values of $Z_1(4, \xi_2)$ and $Z_2(4, \xi_2)$ ($Q = 13$).

| $\xi_2$ | 0 | 1 | 2 | 3 | 4 | 5 | 6 | 7 | 8 | 9 | 10 | 11 | 12 |
|---------|---|---|---|---|---|---|---|---|---|---|----|----|----|
| $\Pr(\xi_2 = l_2)$ | $p_0$ | $p_1$ | $p_2$ | $p_3$ | $p_4$ | $p_5$ | $p_6$ | $p_7$ | $p_8$ | $p_9$ | $p_{10}$ | $p_{11}$ | $p_{12}$ |
| $Z_1(4, \xi_2)$ | $c + 4\theta$ | $c + 4\theta$ | $c + 4\theta$ | $2c + 4\theta$ | $2c + 4\theta$ | $2c + 4\theta$ | $2c + 4\theta$ | $2c + 4\theta$ | $2c + 4\theta$ | $3c + 4\theta$ | $3c + 4\theta$ | $3c + 4\theta$ | $3c + 4\theta$ |
| $Z_2(4, \xi_2)$ | $c$ | $2c$ | $2c$ | $2c$ | $2c$ | $2c$ | $2c$ | $3c$ | $3c$ | $3c$ | $3c$ | $3c$ | $3c$ |

**Table 5.** The distribution of $Z_1(4, \xi_2)$ and $Z_2(4, \xi_2)$ ($Q = 13$).

| $Z_1(4, \xi_2)$ | $c + 4\theta$ | $2c + 4\theta$ | $3c + 4\theta$ | $Z_2(4, \xi_2)$ | $c$ | $2c$ | $3c$ |
|---------|---|---|---|---|---|---|---|
| $\Pr(Z_1(4, \xi_2) = y_1)$ | $\sum_{i=0}^{2} p_i$ | $\sum_{i=3}^{8} p_i$ | $\sum_{i=9}^{12} p_i$ | $\Pr(Z_2(4, \xi_2) = y_2)$ | $p_0$ | $\sum_{i=1}^{6} p_i$ | $\sum_{i=7}^{12} p_i$ |

Because $Z_1(k, \xi_2)$ and $Z_2(k, \xi_2)$ are random variables, and no simple order relation exists between random variables, we cannot solve the problem directly. Therefore, this problem is solved by describing the value of random variables through mathematical expectation. According to Table 2, we transform $E(Z_1(4, \xi_2))$ and $E(Z_2(4, \xi_2))$ into the form of a function

$$E(Z_1(4, \xi_2)) = (c + 4\theta)\sum_{i=0}^{2} p_i + (2c + 4\theta)\sum_{i=3}^{8} p_i + (3c + 4\theta)\sum_{i=9}^{12} p_i.$$
$$E(Z_2(4, \xi_2)) = cp_0 + 2c\sum_{i=1}^{6} p_i + 3c\sum_{i=7}^{12} p_i$$

As in the above analysis, we let $E(Z_1(4, \xi_2)) = E(Z_2(4, \xi_2))$; then, we obtain $c = 4\theta/(\sum_{i=1}^{2} p_i + \sum_{i=7}^{8} p_i)$. Therefore, if $c < 4\theta/(\sum_{i=1}^{2} p_i + \sum_{i=7}^{8} p_i)$, the distribution will occur in the first period; otherwise, the distribution will not occur in the first period. We can see that $\theta$ and $c$ have a positive linear proportional relationship. When $\theta$ is large, $c$ is also large.

In the above analysis, when $Q = 13$, $k = 4$, we obtain the values of $E(Z_1(4, \xi_2))$ and $E(Z_2(4, \xi_2))$, which are convenient for managers in making the correct decisions in the first stage.

Through the above analysis, we can get obtain the 2-stage delay distribution mod-el (model (1)) as follows ($k < Q$):

$$
\begin{cases}
\min z = x_1 c + (1 - x_1)k\theta + \operatorname{int}\left(\frac{\xi_2 + (1 - x_1)k}{Q}\right)c + \delta\left(\frac{\xi_2 + (1 - x_1)k}{Q}\right)c \\
\text{s.t. } x_1 = 0,\ 1.
\end{cases}
\tag{1}
$$

In model (1), $Q$ is the maximum loading capacity of each transport vehicle, and $c$ is the single departure cost of each transport vehicle. $\theta$ is the compensation for the delivery delay in the first stage. If we choose delayed delivery on the first delivery day, $x_1 = 0$, the amount of delayed delivery is $k$ ($k < Q$), and each delayed delivery of goods needs a penalty of $\theta$, otherwise $x_1 = 1$. During the order cycle, all orders must be completed delivered.

### 3.2. 3-Stage Choice Mechanism of Delay Distribution

For the stage that cannot load distribution orders $k$, we assume that the distributions of stages 2 and 3 orders $\xi_2$ and $\xi_3$ are $\Pr(\xi_2 = l_2) = p_{l_2}$, $l_2 = 0, 1, 2, \cdots, m$ and $\Pr(\xi_3 = l_3) = q_{l_3}$, $l_3 = 0, 1, 2, \cdots, n$, where $Z_{11}(k, \xi_2, \xi_3)$ is the distribution cost when we use the delayed distribution in stages 1 and 2; $Z_{12}(k, \xi_2, \xi_3)$ is the distribution cost when we use the distribution strategy in which delivery is delayed in stage 1, and no distribution delay occurs in stage 2; $Z_{21}(k, \xi_2, \xi_3)$ is the distribution cost when we use the distribution strategy in which delivery is delayed in stage 2, and no delay of distribution occurs in stage 1, and $Z_{22}(k, \xi_2, \xi_3)$ is the distribution costs when we do not use the delay distribution in stages 1 and 2. In this work, $Z_{ij}(k, \xi_2, \xi_3)$, $i = 1, 2$, $j = 1, 2$ are random variables, and the specific form is shown below:

$$
Z_{11}(k, \xi_2, \xi_3) =
\begin{cases}
k\theta + (k + \xi_2)\theta + \operatorname{int}\left(\frac{k + \xi_2 + \xi_3}{Q}\right)c + \delta\left(\frac{k + \xi_2 + \xi_3}{Q}\right)c, & k + \xi_2 < Q, \\[2mm]
k\theta + \operatorname{int}\left(\frac{k + \xi_2}{Q}\right)c + \operatorname{int}\left((k + \xi_2) - \operatorname{int}\left(\frac{k + \xi_2}{Q}\right)Q\right)\theta + \\[2mm]
\operatorname{int}\left(\frac{\operatorname{mod}(k + \xi_2, Q) + \xi_3}{Q}\right)c + \delta\left(\frac{\operatorname{mod}(k + \xi_2, Q) + \xi_3}{Q}\right)c, & k + \xi_2 \ge Q,
\end{cases}
$$

$$
Z_{12}(k, \xi_2, \xi_3) = k\theta + \operatorname{int}(k + \xi_2)c + \delta\left(\frac{k + \xi_2}{Q}\right)c + \operatorname{int}\left(\frac{\xi_3}{Q}\right)c + \delta\left(\frac{\xi_3}{Q}\right)c,
$$

$$
Z_{21}(k, \xi_2, \xi_3) =
\begin{cases}
c + \xi_2\theta + \operatorname{int}\left(\frac{\xi_2 + \xi_3}{Q}\right)c + \delta\left(\frac{\xi_2 + \xi_3}{Q}\right)c, & \xi_2 < Q, \\[2mm]
c + \operatorname{int}\left(\frac{\xi_2}{Q}\right)c + \operatorname{int}\left(\xi_2 - \operatorname{int}\left(\frac{\xi_2}{Q}\right)Q\right)\theta + \\[2mm]
\operatorname{int}\left(\frac{\operatorname{mod}(\xi_2, Q) + \xi_3}{Q}\right)c + \delta\left(\frac{\operatorname{mod}(\xi_2, Q) + \xi_3}{Q}\right)c, & \xi_2 \ge Q,
\end{cases}
$$

$$
Z_{22}(k, \xi_2, \xi_3) = c + \operatorname{int}\left(\frac{\xi_2}{Q}\right)c + \delta\left(\frac{\xi_2}{Q}\right)c + \operatorname{int}\left(\frac{\xi_3}{Q}\right)c + \delta\left(\frac{\xi_3}{Q}\right)c.
$$

where $\delta(t) = \begin{cases} 0, & \text{when } t \text{ is an integer}, \\ 1, & \text{other}. \end{cases}$ .

From the above analysis, when only the expected cost is considered, we obtain the following (here, $a_{ij}$, $b_{ij}$ are constants, $i = 1, 2$, $j = 1, 2$):

$$
\begin{aligned}
E(Z_{11}(k, \xi_2, \xi_3)) &= a_{11}\theta + b_{11}c, \\
E(Z_{12}(k, \xi_2, \xi_3)) &= a_{12}\theta + b_{12}c, \\
E(Z_{21}(k, \xi_2, \xi_3)) &= a_{21}\theta + b_{21}c, \\
E(Z_{22}(k, \xi_2, \xi_3)) &= a_{22}\theta + b_{22}c,
\end{aligned}
$$

and when $E(Z_{11}(k, \xi_2, \xi_3)) = \min\{E(Z_{12}(k, \xi_2, \xi_3)), i = 1, 2, j = 1, 2\}$ (that is, the distribution strategy of delaying distribution in stages 1 and 2 is the best one), the following linear inequalities hold:

$$\begin{cases} E(Z_{11}(k, \xi_2, \xi_3)) \le E(Z_{12}(k, \xi_2, \xi_3)), \\ E(Z_{11}(k, \xi_2, \xi_3)) \le E(Z_{21}(k, \xi_2, \xi_3)), \\ E(Z_{11}(k, \xi_2, \xi_3)) \le E(Z_{22}(k, \xi_2, \xi_3)), \end{cases}$$

that is,

$$\begin{cases} a_{11}\theta + b_{11}c \le a_{12}\theta + b_{12}c, \\ a_{11}\theta + b_{11}c \le a_{21}\theta + b_{21}c, \\ a_{11}\theta + b_{11}c \le a_{22}\theta + b_{22}c. \end{cases}$$

In real life, $c \ge 0$, $\theta \ge 0$; therefore, we can see that the region constituted by the above three linear inequalities is convex. The other distribution strategies also result in the above conclusion. Therefore, we obtain the following theorem.

**Theorem 1.** *In the 3-stage delay distribution model, taking the expectation value as the evaluation standard, the dependency region of the single transport cost of each transport vehicle c and the penalty for each car delay in a period-of-time distribution $\theta$ is a convex region.*

**Remark 1.** *In a 2-stage delay distribution model, there is a linear relationship between c and $\theta$, which is analyzed in Section 3.1. When we consider the n-stage delay distribution model ($n \ge 3$), Theorem 2.1 still holds because there is a linear relationship between c and $\theta$. Thus, Theorem 3.1 holds for the n-stage delay distribution model ($n \ge 1$).*

As in the above analysis, we see that no matter if the distribution occurs in each stage, the above conclusion is established. Using the above analysis, we obtain the 3-stage delay distribution as follows:

$$\begin{cases} \min z_3 = \left( \begin{array}{l} x_2\left(x_1 c + (1-x_1)k\theta + \mathrm{int}\left(\frac{\xi + (1-x_1)k}{Q}\right)c + \delta\left(\frac{\xi + (1-x_1)k}{Q}\right)c\right) + (1-x_2)\left[\mathrm{int}\left(\frac{\xi_2 + (1-x_1)k}{Q}\right)c \right. \\ \left. + \delta\left(\frac{\xi_2 + (1-x_1)k}{Q}\right)\theta\right] + \left[\mathrm{int}\left(\frac{\xi_2 + (1-x_1)k}{Q}\right) + \delta\left(\frac{\xi_2 + (1-x_1)k}{Q}\right)\right]c \end{array} \right) \\ \text{s.t. } x_j = 0, 1, j = 1, 2. \end{cases} \tag{2}$$

where $Q$ is the loading capacity of each distribution vehicle. The single transport cost for transporting vehicles is $c$. In each month, if the distribution occurs in the first period, $x_1 = 1$; otherwise, $x_1 = 0$. If the distribution occurs in the second period, $x_2 = 1$; otherwise, $x_2 = 0$. Additionally, $k$ is the part of the first-period demand that cannot be load distribution; $\xi_2$ and $\xi_3$ are the demand of the second and third stages, respectively; $m$ is the total number of distribution vehicles; and $\theta$ is the compensation for distribution delays for a period (ten days). In each month (three periods), the order must be met.

For example, when $Q = 13$, $k = 3$, we consider the delay decision method with 3 stages. At the end of the first stage, the part of the first-stage demand that cannot be a load distribution is 3. The part of the second-stage demand that cannot be a load distribution has 13 possibilities, which are 0, 1, $\cdots$, 12. In each case, we decide between no delivery and delivery of 3 cars in the first stage. In the second stage, in some cases (for example, the part of the second-stage demand that cannot be load distribution is 5), if the 3 cars are not delivered in the first stage, we decide between no delivery and delivery of 8 (3 + 5) cars in the second stage; otherwise, we decide between no delivery and delivery of 5 cars in the second stage. There are 26 possibilities for no delivery or delivery of 3 cars in the first stage. With each possibility, the part of the third-stage demand that cannot be a load distribution also has 13 possibilities, which are 0, 1, $\cdots$, 12. There are 676 ($26 \times 13 \times 2 = 26^2$) possibilities for no delivery or partial delivery of cars in the second stage, and so on. If we consider the $n$-stage delay distribution model, there are $26^n$ possibilities for no delivery or partial delivery of cars in the second stage. From this

analysis, we can see that the computational complexity is too high to solve using analytical methods. Next, we consider the demand as a random variable, with the corresponding expectation and variance.

### 4. The Generalized Expectation Value Model of Delay Distribution in a Random Environment

The goal is to build a model for each time period (including $n$ cycles) that minimizes the overall distribution cost. In this paper, we consider only the two most important costs, namely, the vehicle transportation cost and the delay penalty fee. Using the above analysis, we obtain the $n$-stage delay distribution as follows:

$$
\begin{cases}
\min & z_n = z_{n-1}x_{n-1} + (1-x_{n-1})\left[\mathrm{int}\left(\frac{\eta_{n-1}}{Q}\right)c + \delta\left(\frac{\eta_{n-1}}{Q}\right)\theta\right] + \left[\mathrm{int}\left(\frac{\eta_n}{Q}\right) + \delta\left(\frac{\eta_n}{Q}\right)\right]c \\
\text{s.t.} & \eta_1 = k, \ \eta_{j+1} = \xi_{j+1} + (1-x_j)\eta_j, \ j = 1, 2, 3, \cdots, n-1, \\
& z_2 = x_1 c + (1-x_1)k\theta + \mathrm{int}\left(\frac{\xi_2+(1-x_1)k}{Q}\right)c + \delta\left(\frac{\xi_2+(1-x_1)k}{Q}\right)c, \\
& x_j = 0, 1, \ j = 1, 2, 3, \cdots, n-1.
\end{cases}
\tag{3}
$$

where $Q$ is the loading capacity of each distribution vehicle, and a single transport cost for transporting vehicles is $c$. In each time period, if the distribution occurs in the $j$-th period, $x_j = 1$; otherwise $x_j = 0$. $\theta$ is the compensation for distribution delays for a period. In each month, the order must be met. $k$ is the part of the first-period demand that cannot be a load distribution, $z_j$ is the cost of the $j$-stage delay distribution, and $\xi_j$ is the demand of the $j$-th stage.

Model (3) is a formal description model of the $n$-stage problem. Using model (3), we can utilize the recursive solution method in Section 4, which can be used in the following models (4) and (5).

Random information cannot be compared in the same manner as real numbers. Thus, model (3) is a formal description. The key to solving model (3) is to select a method to compare random information. Generally, we convert stochastic programming into crisp programming. If we consider the random variable expectation to describe the variables, the expectation value model can be expressed as model (4) (the expectation value model).

$$
\begin{cases}
\min & E(z_n) = E(z_{n-1})x_{n-1} + (1-x_{n-1})E\left(\left[\mathrm{int}\left(\frac{\eta_{n-1}}{Q}\right)c + \delta\left(\frac{\eta_{n-1}}{Q}\right)\theta\right] + \left[\mathrm{int}\left(\frac{\eta_n}{Q}\right) + \delta\left(\frac{\eta_n}{Q}\right)\right]c\right) \\
\text{s.t.} & \eta_1 = k, \ \eta_{j+1} = \xi_{j+1} + (1-x_j)\eta_j, \ j = 1, 2, 3, \cdots, n-1, \\
& E(z_2) = x_1 c + (1-x_1)k\theta + E\left(\mathrm{int}\left(\frac{\xi_2+(1-x_1)k}{Q}\right)c + \delta\left(\frac{\xi_2+(1-x_1)k}{Q}\right)c\right), \\
& x_j = 0, 1, \ j = 1, 2, 3, \cdots, n-1.
\end{cases}
\tag{4}
$$

The model is recursive and crisp programming. If we want to obtain the solution of a 3-stage delay distribution model, we first find the solution of a 2-stage delay distribution model. If we want to obtain the solution of a 4-stage delay distribution model, we first find the solution of a 3-stage delay distribution model, and so on. Therefore, if we want to obtain the solution of model (4) (the expectation value model), we first solve the model with $n$-1 stages. The solution complexity of model (4) is far lower than that of model (3), which has $26^n$ possible solutions.

The expected value is beneficial but merely one characteristic of a random variable. There is an essential difference between a random variable and its expected value. There are many inconsistencies between the expected value model and a random decision in the structure. The core inconsistency is that the decision reliability of the model cannot be reflected. In the statistical sense, min $\xi$ meets the reliability $\beta$, min $a$ meets $\Pr(\xi \leq a) \geq \beta$, and

$$
\Pr(\xi \leq a) \geq \beta \Leftrightarrow \Pr\left(Z \triangleq \frac{\xi - E(\xi)}{\sqrt{D(\xi)}} \leq \frac{a - E(\xi)}{\sqrt{D(\xi)}}\right) \geq \beta \Leftrightarrow \frac{a - E(\xi)}{\sqrt{D(\xi)}} \geq Z_\beta \Leftrightarrow a \geq E(\xi) + Z_\beta\sqrt{D(\xi)}.
\tag{*}
$$

Here, $Z_\beta$ is the $\beta$-quantile of $Z = (\xi - E(\xi))/\sqrt{D(\xi)}$ (that is, $\Pr(Z \leq Z_\beta) \geq \beta$, $\Pr(Z > Z_\beta) \leq 1 - \beta$). Thus, if we regard $Z_\beta$ as a constant $C$, $\min a$ can be converted into $\min\left(E(\xi) + C\sqrt{D(\xi)}\right)$.

The following can be easily seen. (1) In (*), $\beta$ should be a larger value (for example, $\beta \in [0.7, 1]$), and when $\beta$ is large, $Z_\beta$ is a non-negative value (for example, when $\xi$ obeys the normal distribution, $Z_{0.75} = 0.68$, $Z_{0.90} = 1.29$, and $Z_{0.95} = 1.65$; when $\xi$ obeys the exponential distribution, $Z_{0.75} = 0.39$, $Z_{0.90} = 1.30$, and $Z_{0.95} = 1.99$; and when $\xi$ obeys the uniform distribution, $Z_{0.75} = 0.87$, $Z_{0.90} = 1.39$, and $Z_{0.95} = 1.56$). This fact indicates that $C$ in $E(\xi) + C\sqrt{D(\xi)}$ should be non-negative (generally, $0 \leq C \leq 2$). (2) If we regard the variance as the credibility measure index of "using the expected value to collect the value of the random variable," $C$ can be interpreted as a penalty coefficient (the effect weakened by uncertainty). (3) When $C = 0$, $E(\xi) + C\sqrt{D(\xi)}$ is the mathematical expectation $E(\xi)$, which implies that $E(\xi) + C\sqrt{D(\xi)}$ can be regarded as the generalized expectation of $\xi$.

According to this analysis, it is a compound quantization model with both the size characteristics and the uncertainty of the value. The measurement model based on model (3) can be generalized into the following model (5) (called the ***generalized expectation value model***, abbreviated as GEM):

$$
\left\{
\begin{array}{ll}
\min & z = E(z_n) + C\sqrt{D(z_n)}, \\
\text{s.t.} & z_n = z_{n-1}x_{n-1} + (1 - x_{n-1})\left[\operatorname{int}\left(\frac{\eta_{n-1}}{Q}\right)c + \delta\left(\frac{\eta_{n-1}}{Q}\right)\theta\right] + \left[\operatorname{int}\left(\frac{\eta_n}{Q}\right) + \delta\left(\frac{\eta_n}{Q}\right)\right]c, \\
& \eta_1 = k, \ \eta_{j+1} = \xi_{j+1} + (1 - x_j)\eta_j, \ j = 1, 2, 3, \cdots, n-1, \\
& z_2 = x_1 c + (1 - x_1)k\theta + \operatorname{int}\left(\frac{\xi_2 + (1-x_1)k}{Q}\right)c + \delta\left(\frac{\xi_2 + (1-x_1)k}{Q}\right)c, \\
& x_j = 0, 1, \ j = 1, 2, 3, \cdots, n-1.
\end{array}
\right.
\tag{5}
$$

Using the above discussion, we can see that (1) when $C = 0$, model (5) is model (4), and the computational complexities of model (5) and model (4) are the same; (2) model (5) has good structural characteristics and can be explained by the different values, which reflect the degree of randomness in the decision-making process; and (3) model (5) defines the statistical significance (i.e., the reliability quantile of $(z_n - E(z_n))/\sqrt{D(z_n)}$). In the actual problem, we can choose the specific value of $C$ via the distribution characteristics of $(z_n - E(z_n))/\sqrt{D(z_n)}$.

From this analysis, we observe that model (5) contains the traditional random processing method. With the different types of synthesis effect functions, the model shows different decision-making approaches, increasing computational complexity.

## 5. Solution Procedure

From model (3), we can see that we use the solution of $z_2$ to obtain $z_3$, we use the solution of $z_3$ to obtain $z_4$, and so on. Because the solution $z_{i+1}$ depends only on the solution $z_i$, the distribution cost sequence $\{z_1, z_2, \cdots, z_n\}$ is a Markov chain.

As seen in this analysis, we can complete the computation of the $n$-stage delay distribution problem with the following $n$-stage delay distribution Algorithm 1.

---

**Algorithm 1** $n$-stage delay distribution algorithm

---

1: Input the initial delay distribution model (3).
2: Given the parameters $c$, $\theta$, $Q$ and $\eta_2 = \xi_2 + (1 - x_2)\eta_1$, $z_2$ is obtained. Here, $\eta_1 = k$.
3: Calculate the probability of $\eta_2$ by $\xi_2$. We obtain the distribution function of $\eta_2$.
4: Through the cycle, the distribution function of $\eta_i$ can be determined by $\xi_i$.
5: Using $z_i$ and $\eta_i = \xi_i + (1 - x_i)\eta_{i-1}$, $z_{i+1}$ is calculated.
6: Select a proper generalized expectation value function $z = E(z_n) + C\sqrt{D(z_n)}$, $C \geq 0$.
7: Convert the stochastic programming model to the GEM.
8: Calculate all possible values of $E(z_n) + C\sqrt{D(z_n)}$, $C \geq 0$ and choose the optimal value to determine the GEM solution.
9: Obtain the solution.

---

**Remark 2.** *If the GEM is a convex function, we can use a convex programming solution method to solve it. If the objective function is linear and the constraints are the selected synthesizing effect*

*function, theGEM is a convex function. However, these are strong conditions, theGEM is commonly not a convex function, and we cannot solve it through traditional methods [38–40].*

### 6. Numerical Examples

In this section, selected examples are presented to demonstrate the effectiveness of the proposed model.

**Case 1** (**2-*stage delay distribution problem***). The automobile manufacturer cooperates with the third-party logistics company, and the third-party logistics company is responsible for transporting the produced automobiles to the automobile dealers. There are two delivery dates in a delivery cycle, that is, two time nodes. If the goods are not delivered in time on the first delivery day, the third-party logistics company needs to compensate the car dealers, and the compensation for each commercial car is $\theta$. All orders need to be completed before the second delivery date.

The third-party logistics company's distribution vehicle load capacity is 12 ($Q = 12$), and the vehicle type is the same, the unit transportation cost of each distribution vehicle is C. The demand on each delivery day is a random variable. If the delivery is timely, the transportation cost is $Z_1(k, \xi_2)$; Otherwise, the transportation cost is $Z_2(k, \xi_2)$.

After the distribution of the commercial vehicles before the first delivery date is arranged according to the load of the distribution vehicles, the remaining 9 vehicles do not meet the full load distribution standard. In order to reduce the cost of the third party logistics enterprises, we arrange according to the distribution demand of the next delivery day. The distribution demand of the next delivery day is predicted by the historical data, as shown in Table 6. (These data are obtained from actual research and statistics and have a certain degree of credibility.)

**Table 6.** Stage-2 demand probability forecast in a month.

| Demand Number $\xi_2$ | 2 | 6 | 7 | 8 | 9 | 10 | 11 | 12 | 13 | 15 |
|---|---|---|---|---|---|---|---|---|---|---|
| $\Pr(\xi_2 = l_2)$ | 0.05 | 0.07 | 0.12 | 0.19 | 0.23 | 0.13 | 0.08 | 0.05 | 0.05 | 0.03 |

The demand in stage 2 is $\xi_2$. With Table 3, we obtain $E(\xi_2) = 8.83$. According to the demand distribution, we obtain the cost when the demand for vehicles is 8 in this period.

We obtain $E(z_1) = 1.95c + 9\theta$ and $E(z_2) = 2.08c$. Using the above analysis, we obtain Tables 8 and 9 in Table 7.

**Table 7.** Total cost in two periods (with the demand for vehicles in this period = 8).

| $\xi_2$ | 2 | 6 | 7 | 8 | 9 | 10 | 11 | 12 | 13 | 15 |
|---|---|---|---|---|---|---|---|---|---|---|
| $\Pr(\xi_2 = l_2)$ | 0.05 | 0.07 | 0.12 | 0.19 | 0.23 | 0.13 | 0.08 | 0.05 | 0.05 | 0.03 |
| $Z_1(k, \xi_2)$ | $c + 9\theta$ | $2c + 9\theta$ | $2c + 9\theta$ | $2c + 9\theta$ | $2c + 9\theta$ | $2c + 9\theta$ | $2c + 9\theta$ | $2c + 9\theta$ | $2c + 9\theta$ | $2c + 9\theta$ |
| $Z_2(k, \xi_2)$ | $2c$ | $2c$ | $2c$ | $2c$ | $2c$ | $2c$ | $2c$ | $2c$ | $3c$ | $3c$ |

**Table 8.** Goods vehicle transport schedule of fees.

| | $\xi_2$ | | 2 | 6 | 7 | 8 | 9 | 10 | 11 | 12 | 13 | 15 | $E(Z_j(k, \xi_2))$ |
|---|---|---|---|---|---|---|---|---|---|---|---|---|---|
| | $\Pr(\xi_2 = i)$ | | 0.05 | 0.07 | 0.12 | 0.19 | 0.23 | 0.13 | 0.08 | 0.05 | 0.05 | 0.03 | |
| | 1 | $Z_1(k, \xi_2)$ | $c + \theta$ | $c + \theta$ | $c + \theta$ | $c + \theta$ | $c + \theta$ | $c + \theta$ | $c + \theta$ | $2c + \theta$ | $2c + \theta$ | $2c + \theta$ | $1.13c + \theta$ |
| | | $Z_2(k, \xi_2)$ | $2c$ | $2c$ | $2c$ | $2c$ | $2c$ | $2c$ | $2c$ | $2c$ | $3c$ | $3c$ | $2.08c$ |
| $k$ | 2 | $Z_1(k, \xi_2)$ | $c + 2\theta$ | $c + 2\theta$ | $c + 2\theta$ | $c + 2\theta$ | $c + 2\theta$ | $c + 2\theta$ | $2c + 2\theta$ | $2c + 2\theta$ | $2c + 2\theta$ | $2c + 2\theta$ | $1.21c + 2\theta$ |
| | | $Z_2(k, \xi_2)$ | $2c$ | $2c$ | $2c$ | $2c$ | $2c$ | $2c$ | $2c$ | $2c$ | $3c$ | $3c$ | $2.08c$ |
| | 3 | $Z_1(k, \xi_2)$ | $c + 3\theta$ | $c + 3\theta$ | $c + 3\theta$ | $c + 3\theta$ | $c + 3\theta$ | $2c + 3\theta$ | $2c + 3\theta$ | $2c + 3\theta$ | $2c + 3\theta$ | $2c + 3\theta$ | $1.21c + 3\theta$ |
| | | $Z_2(k, \xi_2)$ | $2c$ | $2c$ | $2c$ | $2c$ | $2c$ | $2c$ | $2c$ | $2c$ | $3c$ | $3c$ | $2.08c$ |

**Table 8.** *Cont.*

| | $\xi_2$ | | 2 | 6 | 7 | 8 | 9 | 10 | 11 | 12 | 13 | 15 | $E(Z_j(k, \xi_2))$ |
|---|---|---|---|---|---|---|---|---|---|---|---|---|---|
| | Pr($\xi_2 = i$) | | 0.05 | 0.07 | 0.12 | 0.19 | 0.23 | 0.13 | 0.08 | 0.05 | 0.05 | 0.03 | |
| $k$ | 4 | $Z_1(k, \xi_2)$ | $c + 4\theta$ | $c + 4\theta$ | $c + 4\theta$ | $c + 4\theta$ | $2c + 4\theta$ | $2c + 4\theta$ | $2c + 4\theta$ | $2c + 4\theta$ | $2c + 4\theta$ | $2c + 4\theta$ | $1.57c + 4\theta$ |
| | | $Z_2(k, \xi_2)$ | $2c$ | $2c$ | $2c$ | $2c$ | $2c$ | $2c$ | $2c$ | $2c$ | $3c$ | $3c$ | $2.08c$ |
| | 5 | $Z_1(k, \xi_2)$ | $c + 5\theta$ | $c + 5\theta$ | $c + 5\theta$ | $2c + 5\theta$ | $2c + 5\theta$ | $2c + 5\theta$ | $2c + 5\theta$ | $2c + 5\theta$ | $2c + 5\theta$ | $2c + 5\theta$ | $1.76c + 5\theta$ |
| | | $Z_2(k, \xi_2)$ | $2c$ | $2c$ | $2c$ | $2c$ | $2c$ | $2c$ | $2c$ | $2c$ | $3c$ | $3c$ | $2.08c$ |
| | 6 | $Z_1(k, \xi_2)$ | $c + 6\theta$ | $c + 6\theta$ | $2c + 6\theta$ | $2c + 6\theta$ | $2c + 6\theta$ | $2c + 6\theta$ | $2c + 6\theta$ | $2c + 6\theta$ | $2c + 6\theta$ | $2c + 6\theta$ | $1.88c + 6\theta$ |
| | | $Z_2(k, \xi_2)$ | $2c$ | $2c$ | $2c$ | $2c$ | $2c$ | $2c$ | $2c$ | $2c$ | $3c$ | $3c$ | $2.08c$ |
| | 7 | $Z_1(k, \xi_2)$ | $c + 7\theta$ | $2c + 7\theta$ | $2c + 7\theta$ | $2c + 7\theta$ | $2c + 7\theta$ | $2c + 7\theta$ | $2c + 7\theta$ | $2c + 7\theta$ | $2c + 7\theta$ | $2c + 7\theta$ | $1.95c + 7\theta$ |
| | | $Z_2(k, \xi_2)$ | $2c$ | $2c$ | $2c$ | $2c$ | $2c$ | $2c$ | $2c$ | $2c$ | $3c$ | $3c$ | $2.08c$ |
| | 8 | $Z_1(k, \xi_2)$ | $c + 8\theta$ | $2c + 8\theta$ | $2c + 8\theta$ | $2c + 8\theta$ | $2c + 8\theta$ | $2c + 8\theta$ | $2c + 8\theta$ | $2c + 8\theta$ | $2c + 8\theta$ | $2c + 8\theta$ | $1.95c + 8\theta$ |
| | | $Z_2(k, \xi_2)$ | $2c$ | $2c$ | $2c$ | $2c$ | $2c$ | $2c$ | $2c$ | $2c$ | $3c$ | $3c$ | $2.08c$ |
| | 9 | $Z_1(k, \xi_2)$ | $c + 9\theta$ | $2c + 9\theta$ | $2c + 9\theta$ | $2c + 9\theta$ | $2c + 9\theta$ | $2c + 9\theta$ | $2c + 9\theta$ | $2c + 9\theta$ | $2c + 9\theta$ | $2c + 9\theta$ | $1.95c + 9\theta$ |
| | | $Z_2(k, \xi_2)$ | $2c$ | $2c$ | $2c$ | $2c$ | $2c$ | $2c$ | $2c$ | $2c$ | $3c$ | $3c$ | $2.08c$ |
| | 10 | $Z_1(k, \xi_2)$ | $c + 10\theta$ | $2c + 10\theta$ | $2c + 10\theta$ | $2c + 10\theta$ | $2c + 10\theta$ | $2c + 10\theta$ | $2c + 10\theta$ | $2c + 10\theta$ | $2c + 10\theta$ | $3c + 10\theta$ | $1.98c + 10\theta$ |
| | | $Z_2(k, \xi_2)$ | $2c$ | $2c$ | $2c$ | $2c$ | $2c$ | $2c$ | $2c$ | $2c$ | $3c$ | $3c$ | $2.08c$ |
| | 11 | $Z_1(k, \xi_2)$ | $2c + 11\theta$ | $2c + 11\theta$ | $2c + 11\theta$ | $2c + 11\theta$ | $2c + 11\theta$ | $2c + 11\theta$ | $2c + 11\theta$ | $2c + 11\theta$ | $2c + 11\theta$ | $3c + 11\theta$ | $2.03c + 11\theta$ |
| | | $Z_2(k, \xi_2)$ | $2c$ | $2c$ | $2c$ | $2c$ | $2c$ | $2c$ | $2c$ | $2c$ | $3c$ | $3c$ | $2.08c$ |

**Table 9.** Not loaded with average cost.

| $k$ | 1 | 2 | 3 | 4 | 5 | 6 |
|---|---|---|---|---|---|---|
| $E(Z_1(k, \xi_2))$ | $1.13c + \theta$ | $1.21c + 2\theta$ | $1.21c + 3\theta$ | $1.57c + 4\theta$ | $1.76c + 5\theta$ | $1.88c + 6\theta$ |
| $E(Z_2(k, \xi_2))$ | $2.08c$ | $2.08c$ | $2.08c$ | $2.08c$ | $2.08c$ | $2.08c$ |
| $k$ | 7 | 8 | 9 | 10 | 11 | |
| $E(Z_1(k, \xi_2))$ | $1.95c + 7\theta$ | $1.95c + 8\theta$ | $1.95c + 9\theta$ | $1.98c + 10\theta$ | $2.03c + 11\theta$ | |
| $E(Z_2(k, \xi_2))$ | $2.08c$ | $2.08c$ | $2.08c$ | $2.08c$ | $2.08c$ | |

With the expectation value model, we only consider the expectation of $Z_1(k, \xi_2)$ and $Z_2(k, \xi_2)$. From Table 6, we can see that, for a given $c$ and $\theta$, $E(Z_1(k, \xi_2))$ is monotonous and increasing about $k$. Using this analysis method, the solution of model (4) (the expectation value model) is $x_1 = 1$, the optimal global solution. The other cases of average cost are addressed with different $c$ and $\theta$ values. Using the proposed algorithm with Matlab 7.0, we obtain the GEM solutions, as shown in Table 10. We obtain the expected cost $E(Z_2(k, \xi_2))$ and the generalized expectation cost $E(Z_2(k, \xi_2)) + C\sqrt{D(Z_2(k, \xi_2))}$.

**Table 10.** The GEM solutions ($n = 2$).

| $c$ | | 1000 | 1000 | 1000 | 1500 | 1500 | 1500 | 2000 | 2000 | 2000 |
|---|---|---|---|---|---|---|---|---|---|---|
| $\theta$ | | 50 | 150 | 200 | 50 | 150 | 200 | 50 | 150 | 200 |
| $C = 0.1$ | $x$ | 1 | 0 | 0 | 1 | 0 | 0 | 0 | 0 | 0 |
| $C = 0.5$ | $x$ | 1 | 1 | 0 | 0 | 1 | 0 | 1 | 0 | 1 |
| $C = 1$ | $x$ | 1 | 0 | 1 | 1 | 0 | 0 | 0 | 1 | 1 |

When $C = 0$, the GEM degenerates into an expectation value model. From Table 10, we note that the proposed method's solutions contain the solutions $x_1 = 1$ and $x_1 = 0$, which are the solutions of the expectation value model. Thus, the solution is more reliable. All results were obtained using a single core of a 24-core AMD EPYC 7451 CPU running at a base clock frequency of 3.1 GHz. The computation time is 0.0421 s.

Model (5) represents different decision preferences. In the generalized expected value model, when $C$ is small, the third-party logistics company mainly focuses on the expected value of transportation cost. When $C$ is large, the third-party logistics company mainly focuses on the variance of the second stage demand.

**Case 2** (*3-stage delay distribution problem*). This case is similar to **Case 1**, except that each month contains three transportation time points, i.e., the 10th, 20th, and 30th.

The demand in stages 2 and 3 is $\xi_2$ and $\xi_3$, and the distributions are $\Pr(\xi_2 = l_2)$ and $\Pr(\xi_3 = l_3)$, respectively, whcih are shown in Table 11. From Table 7, we find that $E(\xi_2) = 8.92$ and $E(\xi_3) = 9.17$. As in the analysis in Section 3.2 and Theorem 3.1, the relation of $c$ and $\theta$ is as shown in Figure 1.

**Table 11.** Stages 2 and 3 demand distribution ($\Pr(\xi_2 = l_2)$ and $\Pr(\xi_3 = l_3)$) forecast in a month.

| Demand | 2 | 6 | 7 | 8 | 9 | 10 | 11 | 12 | 13 | 15 |
|---|---|---|---|---|---|---|---|---|---|---|
| $\Pr(\xi_2 = l_2)$ | 0.04 | 0.08 | 0.1 | 0.2 | 0.21 | 0.15 | 0.09 | 0.05 | 0.06 | 0.02 |
| $\Pr(\xi_3 = l_3)$ | 0.05 | 0.08 | 0.12 | 0.15 | 0.15 | 0.17 | 0.11 | 0.06 | 0.04 | 0.07 |

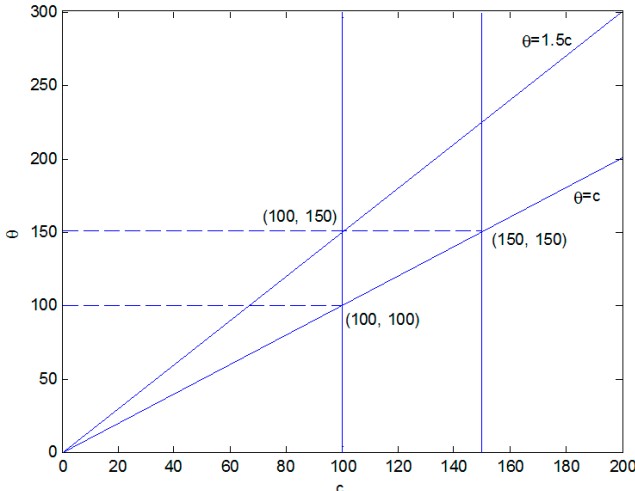

**Figure 1.** The function of $\theta$ on $c$ ($n = 3$).

If we only consider the expectation value, we can observe that, for $c = 100$, when $\theta \leq 0$, we use the delay distribution in stages 1 and 2. When $0 < \theta \leq 100$, we use the distribution strategy to delay delivery in stage 1, with no delay in stage 2. When $100 < \theta \leq 150$, we use the distribution strategy in which delivery is delayed in stage 2, and no distribution delay occurs in stage 1. When $\theta > 150$, we do not use the delay distribution in stages 1 and 2. We can also determine that, for $\theta = 1000$, when $c \leq 0$, we do not use the delay distribution in stages 1 and 2. When $0 < c \leq 1000$, we use the distribution strategy in which delivery is delayed in stage 2 with no distribution delay in stage 1. When $1000 < c \leq 1500$, we use the distribution strategy in which delivery is delayed in stage 1, and there is no distribution delay in stage 2. When $c > 1500$, we use the distribution delay in stages 1 and 2.

Using the proposed algorithm with Matlab 7.0, we obtain the GEM solutions, as shown in Table 12.

From Table 12, we note that when $C = 0.1$, $c = 1500$, $\theta = 1500$, the solution of the proposed method contains the solution $x_1 = 1$, $x_2 = 0$, which is the solution of the expectation value model. With different $C$ values, the solutions are different, which shows different decision-making preferences. When the $C$ value is small, the solution is the same as that of the expectation value model. When the $C$ value is large, the solution indicates that more attention should be paid to the variance in the random variable. Thus, the solution is more reliable. All results were obtained using a single core of a 24-core AMD EPYC 7451 CPU running at a base clock frequency of 3.1 GHz. The computation time is 0.1266 s.

**Table 12.** The GEM solutions ($n = 3$).

|  | $c$ | 100 | 100 | 100 | 500 | 1500 | 2000 |
|---|---|---|---|---|---|---|---|
|  | $\theta$ | 50 | 150 | 200 | 1500 | 1500 | 1500 |
| $C = 0.1$ | $x_1$ | 1 | 0 | 1 | 0 | 1 | 1 |
|  | $x_2$ | 0 | 1 | 1 | 1 | 0 | 1 |
| $C = 0.5$ | $x_1$ | 0 | 0 | 0 | 0 | 1 | 1 |
|  | $x_2$ | 0 | 1 | 1 | 1 | 0 | 0 |
| $C = 1$ | $x_1$ | 0 | 0 | 1 | 0 | 0 | 0 |
|  | $x_2$ | 0 | 0 | 0 | 0 | 1 | 1 |

**Case 3** (*4-stage delay distribution problem*). This case is similar to *Case 1*, except that each month contains four transportation time points, i.e., the 7th, 14th, 21st, and 30$^{\text{th}}$, as shown in Table 13.

**Table 13.** Stages 2, 3 and 4 demand distribution ($\Pr(\xi_2 = i)$, $\Pr(\xi_3 = i)$ and $\Pr(\xi_4 = i)$) forecast in a month.

| Demand | 2 | 6 | 7 | 8 | 9 | 10 | 11 | 12 | 13 | 15 |
|---|---|---|---|---|---|---|---|---|---|---|
| $\Pr(\xi_2 = l_2)$ | 0.04 | 0.08 | 0.1 | 0.2 | 0.21 | 0.15 | 0.09 | 0.05 | 0.06 | 0.02 |
| $\Pr(\xi_3 = l_3)$ | 0.05 | 0.08 | 0.12 | 0.15 | 0.15 | 0.17 | 0.11 | 0.06 | 0.04 | 0.07 |
| $\Pr(\xi_4 = l_4)$ | 0.06 | 0.09 | 0.18 | 0.11 | 0.12 | 0.13 | 0.16 | 0.03 | 0.02 | 0.10 |

The demand in stages 2, 3 and 4 is $\xi_2$, $\xi_3$, $\xi_4$, and the distributions are $\Pr(\xi_2 = l_2)$, $\Pr(\xi_3 = l_3)$, $\Pr(\xi_4 = l_4)$, respectively. From Table 11, we obtain $E(\xi_2) = 8.92$, and $E(\xi_4) = 9.06$. As in the analysis in Section 3.2, we can determine the relation of $c$ and $\theta$, as shown in Figure 2.

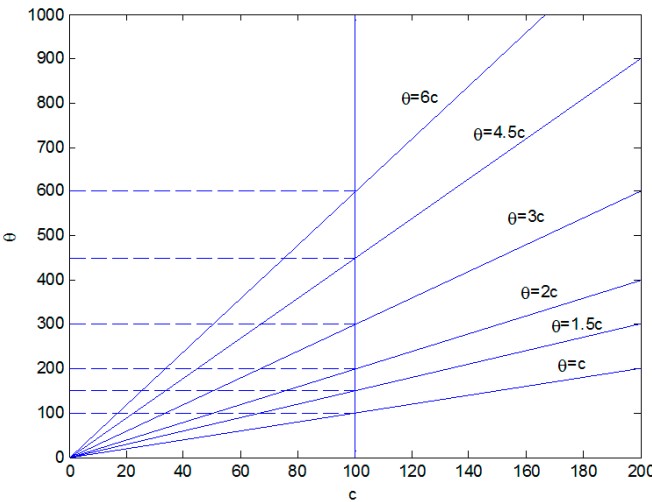

**Figure 2.** The function of $\theta$ on $c$ ($n = 4$).

Next, we only consider, for a given $c$, the changes that range near $\theta$. For a given $\theta$, the change analysis of $c$ is similar. For a given $c = 100$, when $\theta \leq 0$, we use the delay distribution in stages 1, 2, and 3. When $0 < \theta \leq 100$, we use the distribution strategy to delay delivery in stage 1, with no distribution delay in stages 2 and 3. When $100 < \theta \leq 150$, we use the distribution strategy in which delivery is delayed in stage 2 with no distribution delay in stages 1 and 3. When $150 < \theta \leq 200$, we use the distribution strategy in which delivery is delayed in stage 3, and no distribution delay occurs in stages 1 and 2. When

$200 < \theta \le 300$, we use the distribution strategy in which delivery is delayed in stages 1 and 2, and no distribution delay occurs in stage 3. When $300 < \theta \le 450$, we use the distribution strategy in which delivery is delayed in stages 1 and 3, and no distribution delay occurs in stage 2. When $450 < \theta \le 600$, we use the distribution strategy in which delivery is delayed in stages 2 and 3, and no distribution delay occurs in stage 1. When $\theta > 600$, we use the distribution strategy in which delivery is delayed in stages 1, 2, and 3.

Using the proposed algorithm with Matlab 7.0, we obtain the GEM solutions, as shown in Table 14.

**Table 14.** The GEM solutions ($n = 4$).

|  | $c$ | 100 | 100 | 100 | 100 | 100 | 100 | 100 |
|---|---|---|---|---|---|---|---|---|
|  | $\theta$ | 50 | 150 | 200 | 300 | 450 | 600 | 1000 |
| | $x_1$ | 1 | 0 | 0 | 0 | 1 | 1 | 1 |
| $C = 0.1$ | $x_2$ | 0 | 1 | 1 | 1 | 0 | 0 | 1 |
| | $x_3$ | 1 | 1 | 0 | 0 | 1 | 1 | 1 |
| | $x_1$ | 0 | 0 | 0 | 1 | 0 | 0 | 1 |
| $C = 0.5$ | $x_2$ | 0 | 1 | 1 | 1 | 1 | 1 | 1 |
| | $x_3$ | 1 | 1 | 1 | 0 | 1 | 1 | 1 |
| | $x_1$ | 0 | 1 | 0 | 1 | 0 | 1 | 1 |
| $C = 1$ | $x_2$ | 0 | 0 | 1 | 1 | 1 | 1 | 1 |
| | $x_3$ | 1 | 1 | 1 | 0 | 1 | 1 | 1 |

From Table 14, we note that when $C = 0.1$, $c = 100$, $\theta = 150$, the solution of the proposed method contains the solution $x_1 = 0$, $x_2 = 1$, $x_3 = 0$, which is the solution of the expected value model. With different $C$ values, the solutions are different and show different decision-making preferences. When the $C$ value is small, the solution is the same as the expected value model. When the $C$ value is large, the solution pays more attention to the variance of the random variable. Thus, the solution is more reliable. All results were obtained using a single core of a 24-core AMD EPYC 7451 CPU running at a base clock frequency of 3.1 GHz. The computation time is 0.3214 s.

**Case 4** (*10-stage delay distribution problem*). This case is similar to *Case 1*, except that each month contains ten transportation time points, i.e., the 3rd, 6th, 9th, $\cdots$ 30th.

According to historical data, we forecast the next 9 stage node demand probability distributions as shown in Table 15.

**Table 15.** Stage-$n$ demand probability forecast in a month.

| **Demand** | **2** | **6** | **7** | **8** | **9** | **10** | **11** | **12** | **13** | **15** |
|---|---|---|---|---|---|---|---|---|---|---|
| $\Pr(\varsigma_n = i)$. | 0.04 | 0.08 | 0.1 | 0.2 | 0.21 | 0.15 | 0.09 | 0.05 | 0.06 | 0.02 |

From Table 15, we obtain $E(\varsigma_n) = 8.92$, $n = 2, 3, \cdots, 10$. For $C = 0.1$ and $c = 100$, using the proposed algorithm with Matlab 7.0, we obtain the GEM solutions, as shown in Table 16.

From Table 16, we note that when $C = 0.1$, $c = 100$, $\theta = 200$, the solutions of the proposed method contain the solution $x_1 = 0$, $x_2 = 1$, $x_3 = 0$, $x_4 = 0$, $x_5 = 1$, $x_6 = 0$, $x_7 = 0$, $x_8 = 1$, $x_9 = 0$, which is the solution of the expected value model. Thus, the solution is more reliable. All results were obtained using a single core of a 24-core AMD EPYC 7451 CPU running at a base clock frequency of 3.1 GHz.The computation time is 1.2279 s. From *Case 4*, we can see that, if the stage 2 to stage-$n$ node demand probability distributions are given, the proposed method can be developed into an $n$-stage case.

**Table 16.** The GEM solutions ($n = 10$).

| $c$ | 100 | 100 | 100 | 100 | 100 | 100 | 100 | 100 | 100 |
|-----|-----|-----|-----|-----|-----|-----|-----|-----|-----|
| $\theta$ | 50 | 150 | 200 | 300 | 450 | 600 | 800 | 1000 | 2000 |
| $x_1$ | 1 | 0 | 0 | 0 | 1 | 1 | 1 | 0 | 0 |
| $x_2$ | 0 | 1 | 1 | 1 | 0 | 0 | 0 | 1 | 1 |
| $x_3$ | 1 | 0 | 0 | 0 | 1 | 1 | 1 | 0 | 1 |
| $x_4$ | 0 | 0 | 0 | 0 | 0 | 1 | 0 | 1 | 0 |
| $x_5$ | 0 | 1 | 1 | 0 | 1 | 0 | 0 | 1 | 0 |
| $x_6$ | 1 | 1 | 0 | 1 | 1 | 1 | 1 | 0 | 1 |
| $x_7$ | 1 | 0 | 0 | 0 | 0 | 1 | 0 | 0 | 0 |
| $x_8$ | 0 | 0 | 1 | 0 | 0 | 0 | 0 | 1 | 1 |
| $x_9$ | 0 | 1 | 0 | 0 | 1 | 1 | 1 | 0 | 0 |

From the above analysis, we can see that the proposed model can solve the $n$-stage delayed distribution problem. It can be concluded whether the distribution should be delayed at each stage with different $c$ and $\theta$. This conclusion can well assist logistics enterprises to make the optimal decision, to reduce the total cost and improve the efficiency of logistics enterprises. In real life, the stage number is not large, so the proposed model has good system structure features and interpretability, and it can be used in a wide variety of applications.

## 7. Conclusions

In this paper, we consider a class of the $n$-stage delay distribution problem based on the compensation mechanism in a random environment. First, using the recurrence relation, the $n$-stage delay stochastic distribution model is established. Second, with the synthesizing effect function, the stochastic distribution problem is converted into a deterministic GEM, and the proposed model contains different decision-making preferences. Therefore, the solution is more reliable than the expectation model, and the proposed method is more concise and effective. Third, the GEM solutions can contain those of the traditional solution methods. The GEM solution is better than the traditional solution methods if we select the proper synthesis effect functions. Fourth, the dependency region of the single transport cost for each transport vehicle and the penalty for each car delay in a period-of-time distribution is a convex region. Finally, we give a decision-making method for the multi-stage delayed distribution problem. It has a certain guiding significance for optimizing the logistics distribution process.

In practical problems, the multi-stage delay distribution problem is more complex. This paper only discusses making the optimal decision based on given delay compensation. As the amount of compensation increases with the increase of the delay stage, it is not a fixed value. With the limitation of user satisfaction of distribution delay data, this paper does not give the pricing mechanism of phased delay compensation. In the future, we can further discuss the delay compensation pricing mechanism and give the delay compensation strategies in different stages to improve the delay distribution mechanism.

**Author Contributions:** Conceptualization, L.Z.; methodology, Y.Q.; formal analysis, F.L. All authors have read and agreed to the published version of the manuscript.

**Funding:** This research was funded by the National Natural Science Foundation of China (71771078), the Youth Top Talent Project of Research Project of Humanities and Social Sciences in Colleges and Universities of Hebei Province (BJ2021088), the Soft Science Research Project of Innovation Capacity Promotion Program of Hebei Province (21557629D), the Research Project on the Development of Social Sciences in Hebei Province in 2021 (20210201325).

**Institutional Review Board Statement:** Not applicable.

**Informed Consent Statement:** Not applicable.

**Data Availability Statement:** Not applicable.

**Acknowledgments:** This work was supported by the National Natural Science Foundation of China (71771078), the Youth Top Talent Project of Research Project of Humanities and Social Sciences in Colleges and Universities of Hebei Province (BJ2021088), the Soft Science Research Project of Innovation Capacity Promotion Program of Hebei Province (21557629D), Research Project on the Development of Social Sciences in Hebei Province in 2021 (20210201325).

**Conflicts of Interest:** The authors declare no conflict of interest.

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
