# Peer review of "Research on the n-Stage Delay Distribution Method Based on a Compensation Mechanism in a Random Environment"

_axioms, doi:10.3390/axioms11020067_

Round 1

Reviewer 1 Report

The submitted paper addresses the n-stage delay distribution method based on a compensation mechanism in a random environment. Beginning with the 2 and 3-stage delay distribution problem, the characteristics and computational complexity of the model are analysed, and a formal model description of the n-stage problem is given. Combined with specific examples, theoretical analysis and example calculations are demonstrated to show that the formal description model is a good platform for further combinations of solution methods.

The paper addresses an important topic of transportation science. However, there are some concerns about the content of this paper and therefore comments for Authors are given in the following sections.

The first comment is that the status of all used symbols is not completely clear. In this way, Authors are kindly encouraged to present the separate section Notations in order to define and explain:

First subsection of section Notations: Indices and Sets

Second subsection of section Notations: Constants (or Parameters)

Third subsection of section Notations: Decision variables (continuous, integer, binary)

Each index, parameter, variable, etc., contained in the paper should be presented by a unique symbol and only indices with clear domain should be presented as subscripts. An alphabetical order of symbols within each subsection of section Notations would be preferable.

The paper states: on page 10 of 14 ‘‘The computation time is 0.0421 s.’’; on page 11 of 14 ‘‘The computation time is 0.1266 s.’’; on page 12 of 14 ‘‘The computation time is 0.3214 s.’’; on page 13 of 14 ‘‘The computation time is 1.2279 s.’’. Reviewer’s Comment: Authors are kindly encouraged to present:

a/ computer hardware on which the optimization problems were solved;

b/ applied software (including the source citation) for modelling/solving the optimization problems.

Some minor comments related to citations and references are also suggested here:

a/ citations should be presented in accordance with ‘‘Instructions for Authors’’,

b/ references on page 14 of 14 should be ordered and presented in accordance with ‘‘Instructions for Authors’’.

Finally, Authors are kindly encouraged to use the word ‘‘penalty’’ instead of the word ‘‘punishment’’ throughout the paper.

Author Response

Response to Reviewer 1 Comments

Point 1: 

The first comment is that the status of all used symbols is not completely clear. In this way, Authors are kindly encouraged to present the separate section Notations in order to define and explain:

First subsection of section Notations: Indices and Sets

Second subsection of section Notations: Constants (or Parameters)

Third subsection of section Notations: Decision variables (continuous, integer, binary)

Each index, parameter, variable, etc., contained in the paper should be presented by a unique symbol and only indices with clear domain should be presented as subscripts. An alphabetical order of symbols within each subsection of section Notations would be preferable. 

Response 1: A separate section Notations as Section 2 which contains the first subsection of section Notations: Indices and Sets, the second subsection of section Notations: Constants (or Parameters), and the third subsection of section Notations: Decision variables (continuous, integer, binary) in page 3.

Point 2: The paper states: on page 10 of 14 ‘‘The computation time is 0.0421 s.’’; on page 11 of 14 ‘‘The computation time is 0.1266 s.’’; on page 12 of 14 ‘‘The computation time is 0.3214 s.’’; on page 13 of 14 ‘‘The computation time is 1.2279 s.’’. Reviewer’s Comment: Authors are kindly encouraged to present:

a/ computer hardware on which the optimization problems were solved;

b/ applied software (including the source citation) for modelling/solving the optimization problems.

Response 2: The computer hardware on which the optimization problems were solved and the applied software (including the source citation) for modelling/solving the optimization problems are added in each case in page 11-14.

Point 3: Some minor comments related to citations and references are also suggested here:

a/ citations should be presented in accordance with ‘‘Instructions for Authors’’,

b/ references on page 14 of 14 should be ordered and presented in accordance with ‘‘Instructions for Authors’’.

Response 3: The citations and references have modified in accordance with ‘‘Instructions for Authors’’ in page 15-16.

 (All changes have been marked yellow in the manuscript.)

Reviewer 2 Report

The reviewed manuscript is very interesting, above all from the scientific point of view (thus, methods applied are chosen and interpreted correctly). It presents the research on the n-stage delay distribution method based on a compensation mechanism in a random environment. The particular algorithm is proposed and the dependence of a single transport cost of each transport vehicle and the punishment for each car delay in a period-of-time distribution are discussed. Given the abovementioned, the addressed subject is very topical and appropriate for taken into consideration in order to be published in the journal Axioms. However, multiple minor as well as major deficiencies need to be revised prior to the very publication, as follows:

  1. The English grammar needs to be proofread (spelling checks etc.).
  2. The paper structure is not well arranged. Discussion section wherein all the crucial findings are to be summarized in detail. Furthermore, Conclusion section needs to be significantly expanded. Moreover, Abstract does not comply with a required abstract structure as follows: a) brief description of all the manuscript main parts; b) defining the crucial manuscript objective (aim); c) brief outlining the major distinctions of the research conducted in comparison with other analogous previously published studies. In addition, please define the manuscript main objective explicitly.
  3. Elaborated "Literature review" ought to be extended, wherein a substantial number of topic-related references need to be incorporated. In its current state, a list of references is processed poorly and insufficiently. Please take into account even an ensuing list:
    •  (2021). An inductive active filtering method for low-voltage distribution networks. Machines, 9(11). DOI: 10.3390/machines9110258.
    • (2021). Coordinating a decentralized supply chain with capacity cost compensation. RAIRO - Operations Research, 55, S1789-S1802. DOI: 10.1051/ro/2020056.
    • (2020). Using the Operations Research Methods to Address Distribution Tasks at a City Logistics Scale. Transportation Research Procedia, Vol. 44, pp. 348-355. DOI: 10.1016/j.trpro.2020.02.032.
    • (2020). Multi-objective optimization strategy for distribution network considering V2G-enabled electric vehicles in building integrated energy system. Protection and Control of Modern Power Systems, 5(1). DOI: 10.1186/s41601-020-0154-0.
    • (2020). Possible Application of Solver Optimization Module for Solving Single-circuit Transport Problems, LOGI – Scientific Journal on Transport and Logistics, 11(1), pp. 78-87. DOI: 10.2478/logi-2020-0008.
    • (2017). Modeling the distribution network applying the principles of linear programming. 21st International Scientific on Conference Transport Means 2017, Juodkrante; Lithuania; 20-22 September 2017, Code 135093, pp. 73-77. ISSN 1822-296X.
    • (2020). Multi-objective optimization strategy for distribution network considering V2G-enabled electric vehicles in building integrated energy system. Protection and Control of Modern Power Systems, 5(1). DOI: 10.1186/s41601-020-0154-0.
    • (2018). Transportation Factors in the Distribution of Agricultural Produce to Urban Center in Nigeria, LOGI – Scientific Journal on Transport and Logistics, 9(1), pp. 1-10. DOI: 10.2478/logi-2018-0001.
    • and so on.

Author Response

Response to Reviewer 2 Comments

Point 1: The English grammar needs to be proofread (spelling checks etc.)

Response 1: The spelling and grammatical mistakes and incomplete sentences are modified in the paper.

Point 2: The paper structure is not well arranged. Discussion section wherein all the crucial findings are to be summarized in detail. Furthermore, Conclusion section needs to be significantly expanded. Moreover, Abstract does not comply with a required abstract structure as follows: a) brief description of all the manuscript main parts; b) defining the crucial manuscript objective (aim); c) brief outlining the major distinctions of the research conducted in comparison with other analogous previously published studies. In addition, please define the manuscript main objective explicitly.

Response 2: The paper structure is modified. Conclusion and Abstract are improved as your comments.

Now the Conclusion is “In this paper, we consider a class of the n-stage delay distribution problem based on the compensation mechanism in a random environment. First, using the recurrence relation, the n-stage delay stochastic distribution model is established. Second, with the synthesizing effect function, the stochastic distribution problem is converted into a deterministic GEM, and the proposed model contains different decision-making preferences. Therefore, the solution is more reliable than the expectation model, and the proposed method is more concise and effective. Third, the GEM solutions can contain those of the traditional solution methods. The GEM solution is better than those of the traditional solution methods if we select the proper synthesis effect functions. Fourth, the dependency region of the single transport cost for each transport vehicle and the penalty for each car delay in a period-of-time distribution is a convex region. Finally, we give a decision-making method for multi-stage delayed distribution problem. It has certain guiding significance for the optimization of logistics distribution process.”.

Now the Abstract is “With the development of logistics, the delayed distribution problem based on a compensation mechanism has appeared. The core of this problem is to decide whether to delay the distribution at the beginning of each stage and how to compensate the customers if the delay occurs. In this paper, beginning with the 2 and 3-stage delay distribution problem, the characteristics and computational complexity of the model are analyzed, and a formal model description of the n-stage problem is given. The expected value and variance are used as the centralized quantization description strategy for random variables, and the expected value model and the generalized expectation value model for solving the delay distribution problem are given. The solution algorithm is given, and the dependence of the single transport cost of each transport vehicle and the penalty for each car delay in a period-of-time distribution are analyzed. Combined with specific examples, theoretical analysis and example calculations show that the formal description model is a good platform for further combinations of solution methods. This method extends the general delayed distribution problem to multi-stage delayed distribution, which has guiding significance for decision-makers. The proposed model has good system structure features and interpretability and could be used in a wide variety of applications.”

Point 3: Elaborated "Literature review" ought to be extended, wherein a substantial number of topic-related references need to be incorporated. In its current state, a list of references is processed poorly and insufficiently. Please take into account even an ensuing list:

 (2021). An inductive active filtering method for low-voltage distribution networks. Machines, 9(11). DOI: 10.3390/machines9110258.

(2021). Coordinating a decentralized supply chain with capacity cost compensation. RAIRO - Operations Research, 55, S1789-S1802. DOI: 10.1051/ro/2020056.

(2020). Using the Operations Research Methods to Address Distribution Tasks at a City Logistics Scale. Transportation Research Procedia, Vol. 44, pp. 348-355. DOI: 10.1016/j.trpro.2020.02.032.

(2020). Multi-objective optimization strategy for distribution network considering V2G-enabled electric vehicles in building integrated energy system. Protection and Control of Modern Power Systems, 5(1). DOI: 10.1186/s41601-020-0154-0.

(2020). Possible Application of Solver Optimization Module for Solving Single-circuit Transport Problems, LOGI – Scientific Journal on Transport and Logistics, 11(1), pp. 78-87. DOI: 10.2478/logi-2020-0008.

(2017). Modeling the distribution network applying the principles of linear programming. 21st International Scientific on Conference Transport Means 2017, Juodkrante; Lithuania; 20-22 September 2017, Code 135093, pp. 73-77. ISSN 1822-296X.

(2020). Multi-objective optimization strategy for distribution network considering V2G-enabled electric vehicles in building integrated energy system. Protection and Control of Modern Power Systems, 5(1). DOI: 10.1186/s41601-020-0154-0.

(2018). Transportation Factors in the Distribution of Agricultural Produce to Urban Center in Nigeria, LOGI – Scientific Journal on Transport and Logistics, 9(1), pp. 1-10. DOI: 10.2478/logi-2018-0001.

and so on.

Response 3: The "Literature review” have modified, which added the above literature. But the  literature “(2017). Modeling the distribution network applying the principles of linear programming. 21st International Scientific on Conference Transport Means 2017, Juodkrante; Lithuania; 20-22 September 2017, Code 135093, pp. 73-77. ISSN 1822-296X.” can not be found.

 (All changes have been marked yellow in the manuscript.)

Reviewer 3 Report

Paper is written clearly. The list of literature is rather short. The Related works are mentioned in the Introduction section.

The structure of the paper is not mentioned (maybe it could be added into the end of Intro sect.)

I recognized that analysis and background are discussed in Sects 2 and 3, authors' solution for the procedure is in Sect 4 and in Sect 5, the numerical examples are provided. Authors should explain the principle of selected cases - why those three cases were selected for the example section.

Also - how was the procedure validated?

Minor (typographic) observations:

  • l. 37, add space in citation reference Gschwind (2019)
  • l. 141 - extra space
  • add some extra line under the table, e.g. see Table 2, 6, 7, 8
  • p. 6, the box with formula 11 - what (15) stands for here?
  • l. 464 extra full stop at the beginning of the line
  • each page has generated number 1 in the footer

Author Response

Response to Reviewer 3 Comments

Point 1: The structure of the paper is not mentioned (maybe it could be added into the end of Intro sect.)

Response 1:  The structure of the paper is not mentioned in penultimate paragraph in page 2.

Point 2: I recognized that analysis and background are discussed in Sects 2 and 3, authors' solution for the procedure is in Sect 4 and in Sect 5, the numerical examples are provided. Authors should explain the principle of selected cases - why those three cases were selected for the example section.

Also - how was the procedure validated?

Response 2: These data are obtained from actual research and statistics, and have a certain degree of credibility. This sentence is added in page 10.

Point 3: Minor (typographic) observations:

  • 37, add space in citation reference Gschwind (2019)
  • 141 - extra space
  • add some extra line under the table, e.g. see Table 2, 6, 7, 8
  • 6, the box with formula 11 - what (15) stands for here?
  • 464 extra full stop at the beginning of the line

each page has generated number 1 in the footer

(2018). Transportation Factors in the Distribution of Agricultural Produce to Urban Center in Nigeria, LOGI – Scientific Journal on Transport and Logistics, 9(1), pp. 1-10. DOI: 10.2478/logi-2018-0001.

and so on.

Response 3: The above errors are modified.

 (All changes have been marked yellow in the manuscript.)

Round 2

Reviewer 1 Report

Reviewer’s comment

‘‘The first comment is that the status of all used symbols is not completely clear. In this way, Authors are kindly encouraged to present the separate section Notations in order to define and explain:

First subsection of section Notations: Indices and Sets

Second subsection of section Notations: Constants (or Parameters)

Third subsection of section Notations: Decision variables (continuous, integer, binary)

Each index, parameter, variable, etc., contained in the paper should be presented by a unique symbol and only indices with clear domain should be presented as subscripts. An alphabetical order of symbols within each subsection of section Notations would be preferable.’’

was not addressed adequately.

Authors are expected to prepare the section Notations through mentioned subsections in form of alphabetically ordered lists of symbols (i.e. three subsections = three alphabetically ordered lists) where each symbol is first presented and then explained in separate line (i.e. one symbol = one line). Explanation of each symbol should contain: description, domain and unit of measurement. Description for each decision variable should also contain the information whether the variable is continuous, integer or binary.

Please note that this is a major correction and needs to be implemented in full to recommend this paper to be published. Also, it is necessary that the information about the correspondence for this paper should contain Author’s formal institutional e-mail.

Author Response

Point 1: “The first comment is that the status of all used symbols is not completely clear. In this way, Authors are kindly encouraged to present the separate section Notations in order to define and explain:

First subsection of section Notations: Indices and Sets

Second subsection of section Notations: Constants (or Parameters)

Third subsection of section Notations: Decision variables (continuous, integer, binary)

Each index, parameter, variable, etc., contained in the paper should be presented by a unique symbol and only indices with clear domain should be presented as subscripts. An alphabetical order of symbols within each subsection of section Notations would be preferable.’’

was not addressed adequately.

Authors are expected to prepare the section Notations through mentioned subsections in form of alphabetically ordered lists of symbols (i.e. three subsections = three alphabetically ordered lists) where each symbol is first presented and then explained in separate line (i.e. one symbol = one line). Explanation of each symbol should contain: description, domain and unit of measurement.    Description for each decision variable should also contain the information whether the variable is continuous, integer or binary.

Response 1: A separate section Notations as Section 2 which contains the first subsection of section Notations: Indices and Sets, the second subsection of section Notations: Constants (or Parameters), and the third subsection of section Notations: Decision variables (continuous, integer, binary) in page 3-4.

Point 2: It is necessary that the information about the correspondence for this paper should contain Author’s formal institutional e-mail.

Response 2: I am sorry that I (the correspondence author) haven’t the formal institutional e-mail. Now my e-mail is “zhoulei19@126.com”.

 (All changes have been marked yellow in the manuscript.)

Reviewer 2 Report

Thanks a lot to the authors for performing a due revision. Several errors indeed have been revised (e.g., Abstract structure is alright now and English level has truly been improved) now, but multiple flaws remain unsolved. See them as follows:

  • Discussion section, wherein all the major findings are to be summarized in detail, has not been elaborated yet.
  • Conclusion still needs to be significantly expanded.
  • Literature review should still be extended wherein a substantial number of topic-related references need to be incorporated. Current number of 21 for the scientific paper is insufficient (40 at least is appropriate quantity). Please take into consideration even following list:
    • Yu V.F.; Jodiawan P.; Hou M.L.; Gunawan A. Design of a two-echelon freight distribution system in last-mile logistics considering covering locations and occasional drivers. Transportation Research Part E: Logistics and Transportation Review 2021, 154. DOI: 10.1016/j.tre.2021.102461.
    • Stopka O.; Stopkova M.; Kampf R. Application of the Operational Research Method to Determine the Optimum Transport Collection Cycle of Municipal Waste in a Predesignated Urban Area. Sustainability 11(8), 2019, Article no: 2275. DOI: 10.3390/su11082275.
    • Mancini S.; Gansterer M. Vehicle routing with private and shared delivery locations. Computers and Operations Research 2021, 133. DOI: 10.1016/j.cor.2021.105361.
    • and so forth.
  • Furthermore, added references as follows from the previous revision are indicated improperly (i.e., their names are in wrong form): 
    • "Ondrej S.; Karel J.; Mária S. Using the operations research methods to address distribution tasks at a city logistics scale. Transportation Research Procedia. 2020, 44, 348-355."
    • "Josef, Š.; JiÅ™í ÄŒ.; Mykola G. Possible application of solver optimization module for solving single-circuit transport problems. LOGI-Scientific Journal on Transport and Logistics. 2020, 11, 78-87."
    • "Michael S.; Timo G.; Daniele V. Advances in Vehicle Routing and Logistics Optimization: Exact Methods. EURO Journal on Transportation and Logistics. 2019, 8, 117-118."
    • "Gizem O.; Martin S. An iterative re-optimization framework for the dynamic vehicle routing problem with roaming delivery locations. Transportation Research Part B: Methodological. 2019, 128, 207-235."

whereby the proper forms are as follows:

    • "Stopka O.; JeÅ™ábek K.; Stopková M. Using the operations research methods to address distribution tasks at a city logistics scale. Transportation Research Procedia. 2020, 44, 348-355."
    • "Šedivý J.; ÄŒejka J.; Guchenko M. Possible application of solver optimization module for solving single-circuit transport problems. LOGI-Scientific Journal on Transport and Logistics. 2020, 11, 78-87."
    • "Schneider M.; Gschwind T.; Vigo D. Advances in Vehicle Routing and Logistics Optimization: Exact Methods. EURO Journal on Transportation and Logistics. 2019, 8, 117-118."
    • "Ozbaygin G.; Savelsbergh M. An iterative re-optimization framework for the dynamic vehicle routing problem with roaming delivery locations. Transportation Research Part B: Methodological. 2019, 128, 207-235."

Therefore, I highly recommend the authors revising all the shortcomings once again.

Author Response

Point 1: Discussion section, wherein all the major findings are to be summarized in detail, has not been elaborated yet.

Response 1: All the major findings are to be summarized in detail in page 2-3.

Point 2: Conclusion still needs to be significantly expanded.

Response 2: The conclusion has been significantly expanded.

Now the Conclusion is “In this paper, we consider a class of the n-stage delay distribution problem based on the compensation mechanism in a random environment. First, using the recurrence relation, the n-stage delay stochastic distribution model is established. Second, with the synthesizing effect function, the stochastic distribution problem is converted into a deterministic GEM, and the proposed model contains different decision-making preferences. Therefore, the solution is more reliable than the expectation model, and the proposed method is more concise and effective. Third, the GEM solutions can contain those of the traditional solution methods. The GEM solution is better than those of the traditional solution methods if we select the proper synthesis effect functions. Fourth, the dependency region of the single transport cost for each transport vehicle and the penalty for each car delay in a period-of-time distribution is a convex region. Finally, we give a decision-making method for multi-stage delayed distribution problem. It has certain guiding significance for the optimization of logistics distribution process.

There are a large number of multi-stage problems in real life. Since 2 and 3-stage is the most common, the delayed distribution problem can also be discussed in 3-stage. In 3-stage delayed distribution problem, there are two decision points, which are at the end of the first stage and the end of the second stage (Here, the logistics company will determine whether to distribute). When the first stage is a surplus, the logistics company will priority distribute the rest of the first stage at the end of the second stage, so that customers who want to get the commodity in first stage will wait for only one stage and get a certain compensation. But towards the end of the second phase if the logistics company does not meet the requirements of vehicle distribution, the distribution may be delayed again and customers will wait for the two stages, so compensation should be higher. Therefore, it is also a question worth to discussing whether to increase the compensation according to the two-stage compensation and to discuss the compensation according to the multi-stage compensation.”.

Point 3: Literature review should still be extended wherein a substantial number of topic-related references need to be incorporated. Current number of 21 for the scientific paper is insufficient (40 at least is appropriate quantity). Please take into consideration even following list:

Yu V.F.; Jodiawan P.; Hou M.L.; Gunawan A. Design of a two-echelon freight distribution system in last-mile logistics considering covering locations and occasional drivers. Transportation Research Part E: Logistics and Transportation Review 2021, 154. DOI: 10.1016/j.tre.2021.102461.

Stopka O.; Stopkova M.; Kampf R. Application of the Operational Research Method to Determine the Optimum Transport Collection Cycle of Municipal Waste in a Predesignated Urban Area. Sustainability 11(8), 2019, Article no: 2275. DOI: 10.3390/su11082275.

Mancini S.; Gansterer M. Vehicle routing with private and shared delivery locations. Computers and Operations Research 2021, 133. DOI: 10.1016/j.cor.2021.105361.

and so forth.

Furthermore, added references as follows from the previous revision are indicated improperly (i.e., their names are in wrong form): 

"Ondrej S.; Karel J.; Mária S. Using the operations research methods to address distribution tasks at a city logistics scale. Transportation Research Procedia. 2020, 44, 348-355."

"Josef, Š.; JiÅ™í ÄŒ.; Mykola G. Possible application of solver optimization module for solving single-circuit transport problems. LOGI-Scientific Journal on Transport and Logistics. 2020, 11, 78-87."

"Michael S.; Timo G.; Daniele V. Advances in Vehicle Routing and Logistics Optimization: Exact Methods. EURO Journal on Transportation and Logistics. 2019, 8, 117-118."

"Gizem O.; Martin S. An iterative re-optimization framework for the dynamic vehicle routing problem with roaming delivery locations. Transportation Research Part B: Methodological. 2019, 128, 207-235."

whereby the proper forms are as follows:

"Stopka O.; JeÅ™ábek K.; Stopková M. Using the operations research methods to address distribution tasks at a city logistics scale. Transportation Research Procedia. 2020, 44, 348-355."

"Šedivý J.; ÄŒejka J.; Guchenko M. Possible application of solver optimization module for solving single-circuit transport problems. LOGI-Scientific Journal on Transport and Logistics. 2020, 11, 78-87."

"Schneider M.; Gschwind T.; Vigo D. Advances in Vehicle Routing and Logistics Optimization: Exact Methods. EURO Journal on Transportation and Logistics. 2019, 8, 117-118."

"Ozbaygin G.; Savelsbergh M. An iterative re-optimization framework for the dynamic vehicle routing problem with roaming delivery locations. Transportation Research Part B: Methodological. 2019, 128, 207-235."

Response 3: The "Literature review” have modified and added the above literature. Now the number of literature is 40.

 (All changes have been marked yellow in the manuscript.)

Round 3

Reviewer 1 Report

Reviewer’s comment ‘‘Also, it is necessary that the information about the correspondence for this paper should contain Author’s formal institutional e-mail.’’ needs to be implemented in full to recommend this paper to be published. Formal institutional e-mail of the corresponding Author is necessary (corresponding Author can be changed if the current one possesses no formal institutional e-mail). Correspondence: lifachao@tsinghua.org.cn given in the first version of submitted paper seems acceptable.

Author Response

Point 1: ‘‘Also, it is necessary that the information about the correspondence for this paper should contain Author’s formal institutional e-mail.’’ needs to be implemented in full to recommend this paper to be published. Formal institutional e-mail of the corresponding Author is necessary (corresponding Author can be changed if the current one possesses no formal institutional e-mail). Correspondence: lifachao@tsinghua.org.cn given in the first version of submitted paper seems acceptable.

Response 1: The corresponding Author has been changed to Fachao Li. His e-mail is "lifachao@tsinghua.org.cn”.

 (All changes have been marked yellow in the manuscript.)

Reviewer 2 Report

The authors have revised the original manuscript version truly profoundly, however several of my previous remarks have been disregarded. These are as follows:

  • Discussion section has not been elaborated yet. Discussion ought to be processed prior to the Conclusion, and must encompass and describe in detail all the crucial results obtained.
  • Conclusion still needs to be extended. The authors have omitted to present (analyse) the particular recommendations for further research in the addressed topic as well as certain limitations that have impeded or may impede (jeopardize) some specific aspects under investigation.

Author Response

Point 1: Discussion section has not been elaborated yet. Discussion ought to be processed prior to the Conclusion, and must encompass and describe in detail all the crucial results obtained.

Response 1: The discussion section has been added in page 15. The discussion section is “From the above analysis, we can see that, the proposed model can solve the n-stage delayed distribution problem. It can be concluded whether the distribution should be delayed at each stage with different c and θ. This conclusion can well assist logistics enterprises to make the optimal decision, so as to reduce the total cost and improve the efficiency of logistics enterprises. In real life, the stage number is not large, so the proposed model has good system structure features and interpretability, and it can be used in a wide variety of applications.”

Point 2: Conclusion still needs to be extended. The authors have omitted to present (analyse) the particular recommendations for further research in the addressed topic as well as certain limitations that have impeded or may impede (jeopardize) some specific aspects under investigation.

Response 2: The conclusion has been significantly expanded.

Now the Conclusion is “In this paper, we consider a class of the n-stage delay distribution problem based on the compensation mechanism in a random environment. First, using the recurrence relation, the n-stage delay stochastic distribution model is established. Second, with the synthesizing effect function, the stochastic distribution problem is converted into a deterministic GEM, and the proposed model contains different decision-making preferences. Therefore, the solution is more reliable than the expectation model, and the proposed method is more concise and effective. Third, the GEM solutions can contain those of the traditional solution methods. The GEM solution is better than those of the traditional solution methods if we select the proper synthesis effect functions. Fourth, the dependency region of the single transport cost for each transport vehicle and the penalty for each car delay in a period-of-time distribution is a convex region. Finally, we give a decision-making method for multi-stage delayed distribution problem. It has certain guiding significance for the optimization of logistics distribution process.

In practical problems, the multi-stage delay distribution problem is more complex. This paper only discusses how to make the optimal decision based on given delay compensation. As the amount of compensation increases with the increase of delay stage, it is not a fixed value. With the limitation of user satisfaction of distribution delay data, this paper does not give the pricing mechanism of phased delay compensation. In the future, we can further discuss the delay compensation pricing mechanism and give the delay compensation strategies in different stages to improve the delay distribution mechanism.

 (All changes have been marked yellow in the manuscript.)